

# Dynamic Spectra of Small-Mass Meteors

Emma R. Mirizio[1], Robert G. Michell[2], and Marilia Samara[2]

[1]University of Maryland, College Park, MD, USA
[2]NASA Goddard Space Flight Center, Greenbelt, MD, USA

**Correspondence:** Emma R. Mirizio (emmamirizio@gmail.com)

**Abstract.** We present dynamic (22 frames per second) observations of optical spectra of small-mass (2-200 mg) meteors observed with an EMCCD imager equipped with a diffraction grating. This observational campaign occurred at Arecibo, Puerto Rico during May 2012, resulting in eight hours of clear data over four nights. We detected 22 meteors with this setup and their spectra showed varying compositions, including evidence of Na, Mg, Fe, and Ca. Spectral lines persisting over multiple frames
(up to 23 frames) with sufficient signal, showed evidence for differential ablation. Brighter, more massive meteors had stronger and varied spectral signals, which showed that the temporal and spectral resolution of the faintest meteors approached the noise level of the camera system. Optical and spectral detections of these small-mass meteors provide a greater understanding of the composition of the milligram-sized population of meteors.

**1 Introduction**

Small-mass meteoroids and cosmic dust particles are deposited into Earth's atmosphere when their orbits intersect Earth. This material is a direct result of collisions in the asteroid belt and the sublimation of materials in comets as they approach the sun. When these meteoroids ablate in the upper atmosphere, they deposit various metals which can often coalesce into distinct metal layers within the atmosphere (Vondrak et al., 2008). Various detection methods have been used to place the total amount
of all meteoric material entering the atmosphere at a range from just 5 tons up to 300 tons of material per day (Plane, 2012). This vast discrepancy has drastic implications on the composition of the upper atmosphere. First, the composition of metal and ion layers in the mesosphere (altitudes of 80 to 105 km) is determined by the total amount of material that ablates and ionizes to create these structures. In addition, particles and metals in the atmosphere can be used as markers for climate change (Plane, 2012). A better understanding of the total amount and composition of meteoric material ionized in the atmosphere is needed in
order to better constrain global circulation models of Earth's atmosphere.

Meteor mass estimates are found by two main methods—radar observations and optical observations. In general, radar observations are best at detecting the more numerous smaller mass meteors (Mathews et al., 2001; Close et al., 2005; Dyrud and Janches, 2008; Fentzke and Janches, 2008; Close et al., 2012; Janches et al., 2014b; Michell et al., 2019), while optical observations are better suited to observing the less numerous, larger mass (optically brighter) meteor population. Both of these





methods rely on assumptions about the energy conversion of the meteor during the ablation process in order to estimate the initial meteoroid mass from observable parameters. Radar observations use an assumption about the ionization efficiency, which is essentially the number of free electrons generated per initial meteoric atom. (Campbell-Brown et al., 2012) gives a more complete description of ionization efficiencies including how they are modeled and estimated. Optical observations rely on an assumption about how much of the meteors initial kinetic energy gets converted into photon energy, which is called the

luminous efficiency (McKinley, 1961; Campbell-Brown et al., 2012).

Radar observations to find meteor head-echoes are performed using High-Power and Large Aperture (HPLA) radar, also called Incoherent Scatter Radar (ISR). These are used to determine the flux of a large number of milligram and microgram sized meteors and dust particles, for example (Mathews et al., 2001; Close et al., 2005; Dyrud and Janches, 2008; Fentzke and Janches, 2008; Close et al., 2012; Janches et al., 2014b). This method can be used to determine the line-of-sight velocity

as well as the deceleration along the meteor's path, as well as the backscattered power from the meteor. However, it can only be used to measure a lower limit for the total velocity unless the radar is using interferometry (Sato et al., 2000; Chau and Woodman, 2004; Sparks et al., 2010; Janches et al., 2014a).

Optical observations of the sporadic meteor background typically detect larger and brighter meteors in the milligram to gram range, but sacrificing temporal or spatial resolution allows this detection method to observe very faint meteors. The

combination of optical and radar data provides a more accurate constraint on meteoric properties such as the total velocity and originating radiant direction. Estimates of meteoroid mass have been made by this method as shown in (Campbell-Brown et al., 2012; Weryk and Brown, 2013; Brown et al., 2017; Michell et al., 2019).

Optical spectroscopy can be used to determine the composition of meteors which could enable even better mass estimates to be made, because the ablation process and composition would be better understood and therefore less assumptions would

need to be made in order to calculate the meteoroid mass. Spectroscopic observations of meteors have been thoroughly studied for larger mass meteors, or fireballs (Ceplecha, 1971; Borovicka, 1993; Trigo-Rodriguez et al., 2003), and in meteor showers (Millman, 1963). In addition, all-sky cameras have similarly been used to find greater numbers of fireballs and their spectra (Rudawska et al., 2016). Meteor composition for small mass meteors have been estimated from meteor head-echoes for microgram sized meteors in the sporadic background (Janches et al., 2009). (Jenniskens, 2004) explored the detection and

composition of milligram sized fast meteors within the Leonid shower using an aircraft with attached spectrographs and optical cameras to detect milligram sized meteors.

Small-mass meteors in the sporadic background are more difficult to study optically, because they have faint optical signals, requiring telescopic (small) fields of view and thus the meteors occur less frequently within the imager FOV, requiring longer observation periods. Meteor showers have the advantage that a larger number of meteors can occur in a relatively short time,

thus requiring shorter overall observing times. Spectroscopy can give insight into relative abundances of each element within a meteor, and be used to detect differential ablation when a meteor is observed over a period of time. Common elements in sporadic meteors include Mg, Fe, Ni, and Na as well as smaller abundances of Ca (Harvey, 1973). Ablation of small-mass meteors and dust particles has been modeled to confirm differential ablation and relative abundances of ionized particles in the meteors. More volatile elements, Na and K, ablate earlier, or higher in the atmosphere compared to less volatile elements, Fe,



Mg, and Si, which are all major constituents of meteor composition (Vondrak et al., 2008). Data from the Arecibo radar has
shown evidence of this differential ablation process based on meteor altitude and velocity, for example (Janches et al., 2009).

   The next section describes the overall methodology used in the following analyses. Then a description of the results and
their implications is presented, followed by an overall summary.

## 2 Methodology

### 2.1 Optical


The observations presented here are of meteor spectra taken with a small field of view (13°) Electron Multiplying CCD (EM-
CCD) imager, where meteor spectra were observed for meteors with absolute visual magnitudes down to +3.0, corresponding
to masses down to a few milligrams. These observations were taken during an observational campaign conducted at the Arecibo
Observatory in Puerto Rico (18.3° N, 66.8° W) during May 2012. This campaign utilized an Andor Ixon DU-888 EMCCD
imager that is part of the Multi-spectral Observatory Of Sensitive EMCCDs (MOOSE) imagers, as described in (Michell et al.,
2019). To reduce noise and ensure a signal from faint meteors, the CCD was cooled to -70 C and binned 2 x 2. A short in-
tegration time was chosen, which gave 22 frames per second. The 380 x 380 pixel frame resulted in an angular resolution of
0.036 degrees per pixel, resulting in a total field-of-view of 13 degrees. The optical camera was pointed in the local zenith at
the Arecibo optical site. This campaign produced a total of 8 hours of observations with clear sky without clouds over four
nights from May 17-20, 2012 and a total of 22 meteors were detected with measurable spectra.

   The diffraction grating used produced a spectral resolution of ∼7 nm per pixel and covered the range from 400 nm to 800
nm. The spectral calibration was done in the lab prior to the campaign using a Neon-Argon light source with known spectral
lines. The diffraction grating was aligned on the imager lens such that the spectra were dispersed in the north-south direction
in order to maximize the spectral resolution of the generally east-west traveling meteors.

These observations were taken at the same time as the dual radar and optical observations presented in (Michell et al., 2019).
The motivation was to be able to increase the accuracy of our mass estimates by measuring the actual spectra of the meteors
detected by multiple methods. However, given the reduced sensitivity of the imager with the spectrograph grating and the
larger field-of-view, the meteors that were detected with measurable spectra did not have corresponding radar returns because
they were either too far away from the radar beams, or passed through the nulls in the beam pattern, while still being inside
the imager field-of-view. Therefore no radar data is presented and the focus of this paper is on the properties derived from the
optical spectra alone, including finding lower limits on their velocity and upper limits on mass for each meteor as well as their
elemental compositions.

   Figure 1 shows an example frame from the imager, illustrating a meteor and its spectra (left) and an example image showing
stars and their corresponding spectra (right) that were used as calibration sources. The F0 star shown was used for calibrating
the wavelength sensitivity of the combined imager–diffraction grating system.





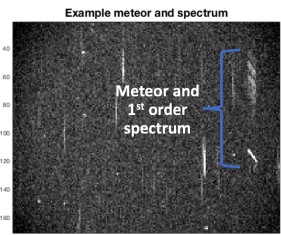 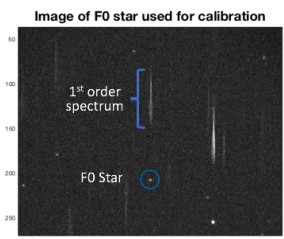

**Figure 1.** An example raw image of a meteor and its spectra (left) that is uncalibrated and unflattened. An example stellar spectra (right) for a type F0 star is shown. This star was used for calibrating the wavelength sensitivity of the combined imager–diffraction grating system.

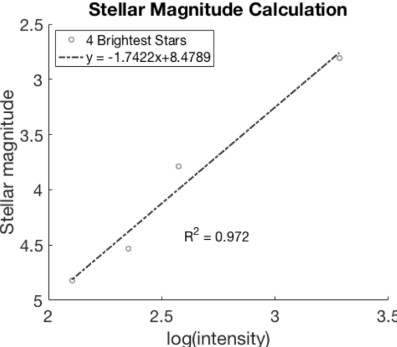

**Figure 2.** The calibration between the stellar magnitudes and the logarithm of the imager counts. The linear fit values and the $R^2$ values are given in the legend.

### 2.2 Stellar Calibration

The first step in the imager calibration was to use reference stars in the images to convert the intensities from the imager (in counts) into stellar magnitudes. This was accomplished by creating a linear fit between the magnitudes of known reference stars and the logarithm of the intensities (in counts) from the imager. Figure 2 shows this relationship for the four brightest

stars in the image, where clear intensities were able to be derived from the zeroth-order spectra (un-diffracted images). For these data, we found a linear fit, with an $r^2$ value of 0.972 using the four brightest stars in an image. This linear fit was then used to convert the imager counts corresponding to the meteors into stellar magnitude values which were then used to calculate the meteor masses, using the kinetic energy and luminosity relationship found in (McKinley, 1961). For the 22 meteors detected, we can place lower bounds on the mass, because we assumed that the only measured component of the velocity (the horizontal

component) was the total speed, knowing the the total speed must have been higher. In addition, without radar returns or another imager for triangulation, we simply assumed that the altitude of all the meteors was 95 km, which will also contribute to the uncertainties in the mass estimates.

In order to characterize the relative intensities of the observed emission lines in the meteor spectra, we need to quantify the imager response with the diffraction grating as a function of wavelength (the spectral response), which was not done in the lab

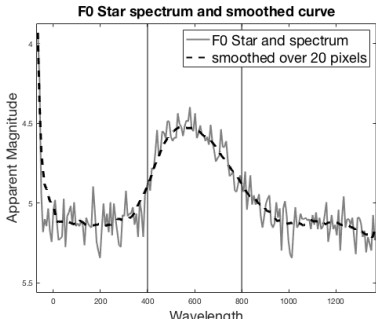
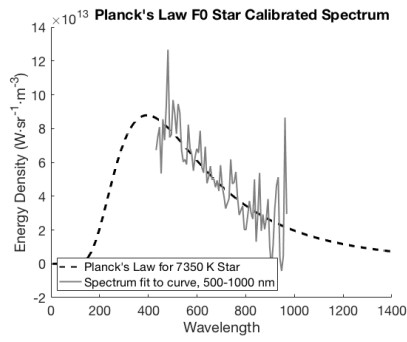

**Figure 3.** The measured F0 stellar spectrum showed that the spectrum responded between 400 and 800 nm (left) and this response was fit to a Planck curve with a temperature of 7350 K (right).

at the time. Due to quantum efficiency effects of the CCD detector itself and the transmission characteristics of all the lenses, there will be a spectral dependence on the measured intensity of the meteor spectra that needs to be normalized out. To do this, a star of a known spectral type (F0 in this case), was examined. Figure 3 shows the measured spectra from an F0 type star, fit to a Planck curve with a temperature of 7350 K. This was then used to create a wavelength normalization that was applied to the meteor spectra, such that the relative intensities of the observed emission lines were of physical significance and not

just instrumentally created. Atmospheric transmittance corrections were also applied to each spectrum that affected the shorter wavelengths (350 - 400 nm).

### 3  Observations

There were a total of 22 meteors observed, where spectra were discernible. Table 1 lists a summary of the main characteristics of these 22 meteors, including the date, time in UT, the horizontal velocity in km/s, the azimuthal direction of origin in degrees

where East = $0°$ and North = $90°$, the maximum brightness in visual magnitude and the derived mass in milligrams. The maximum brightnesses ranged between 3.2 and 0.4 magnitude and the masses ranged between 2.4 and 153.1 mg.

As shown in Table 1, strong spectral lines were visible for Ca, Fe, Mg, and Na in some meteors. Lines were observed at 518 nm and 553 nm for Mg (Savage and Boitnott, 1971), and in all cases where Mg was present, both spectral lines were visible. In addition, strong Na lines were only visible in the brightest meteors that also showed evidence for the other three observed

elements. This analysis was done by observing the full evolution of each spectrum, and peaks were made clear by summing over multiple frames.

Figures 4 and 5 show the spectrum of meteor 4. This meteor was visible for 11 frames, or 0.5 seconds, and higher wavelengths of the spectrum were not visible for the first half of the meteor's path across the detector. The Fe line is not visible until more than halfway through the meteor's path, while Mg is seen as an excited line throughout the spectrum. Although part

of the spectrum is lost due to the limited field of view, the 518 line of Mg is seen in both the beginning and end of the spectrum.

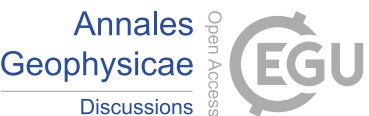
**Table 1.** Summary of the main characteristics of the 22 meteors detected, showing the meteor number, the date, time in UT, the horizontal velocity in km/s, the azimuthal direction of origin in degrees where East = 0° and North = 90°, the maximum brightness in visual magnitude, the derived mass in milligrams, and spectral lines that were discernible.

| # | Date | Time (UT) | H. Vel. (km/s) | Azimuth (deg.) | Mag. | Mass (mg) | Elements Found |
|---|------|-----------|----------------|----------------|------|-----------|----------------|
| 1 | 17 May | 07:18:24 | 39.6 | 101.0° | 2.3 | 4.76 | Ca,Fe |
| 2 | 17 May | 07:20:24 | 35.3 | 151.1° | 2.7 | 5.35 | Ca |
| 3 | 17 May | 07:25:06 | 35.9 | 109.2° | 2.8 | 4.04 | Ca,Fe |
| 4 | 17 May | 07:25:16 | 40.0 | 145.4° | 2.0 | 12.64 | Ca,Mg,Fe |
| 5 | 17 May | 08:16:01 | 32.2 | 151.9° | 2.9 | 2.42 | Ca,Mg,Na |
| 6 | 17 May | 08:20:06 | 25.3 | 167.0° | 2.9 | 7.72 | |
| 7 | 17 May | 08:31:00 | 23.4 | 236.3° | 2.7 | 46.30 | Ca,Mg,Fe,Na |
| 8 | 17 May | 08:37:30 | 19.3 | 121.6° | 2.8 | 12.97 | Ca,Mg,Fe |
| 9 | 17 May | 08:48:25 | 9.0 | 127.4° | 3.2 | 53.09 | |
| 10 | 17 May | 08:49:54 | 38.0 | 109.9° | 2.7 | 7.11 | Ca |
| 11 | 18 May | 05:54:06 | 20.1 | 221.0° | 3.0 | 11.81 | |
| 12 | 18 May | 06:12:18 | 17.9 | 239.8° | 1.8 | 153.16 | Ca,Mg,Fe,Na |
| 13 | 18 May | 06:53:51 | 55.3 | 174.2° | 1.2 | 11.65 | |
| 14 | 19 May | 01:35:59 | 38.1 | 151.2° | 1.9 | 23.43 | |
| 15 | 19 May | 01:49:39 | 24.3 | 146.5° | 1.1 | 84.08 | |
| 16 | 19 May | 08:14:23 | 68.0 | 98.4° | 1.7 | 3.94 | |
| 17 | 19 May | 08:18:10 | 33.6 | 138.2° | 2.0 | 7.64 | |
| 18 | 20 May | 07:41:50 | 46.6 | 52.3° | 1.8 | 10.96 | Mg,Fe,Na |
| 19 | 20 May | 07:55:30 | 23.0 | 346.0° | 2.1 | 55.66 | Ca,Mg |
| 20 | 20 May | 08:04:37 | 62.4 | 107.9° | 1.4 | 5.50 | Ca,Mg,Na |
| 21 | 20 May | 08:33:13 | 77.0 | 154.1° | 1.9 | 4.29 | Mg,Na |
| 22 | 20 May | 08:38:46 | 62.7 | 123.9° | 0.4 | 17.57 | Ca,Fe,Na |

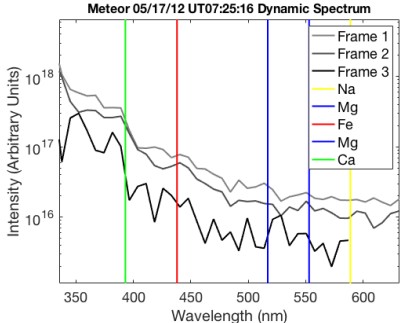

**Figure 4.** Evolution of a meteor over 6 frames, or 0.3 seconds, showing the Fe spectral line appearing after the Mg line remains visible.

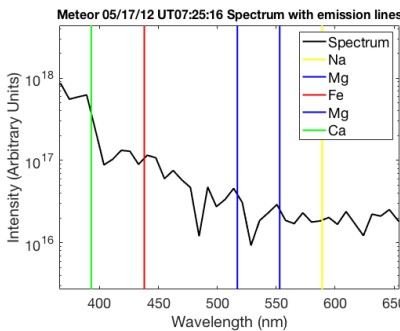

**Figure 5.** Meteor showing evidence of all elements listed except Na.

## 4   Discussion and Conclusions

Differential ablation has previously been detected at Arecibo using the 430 MHz radar by observing the head-echoes of small-mass meteors (Janches et al., 2009) and similarly detected by the Resolute Bay ISR, another HPLA radar (Malhotra and Mathews, 2011). With radar, compositional differences were determined by measurements made at various altitudes over time.

At higher altitudes, more volatile elements are observed, confirming that the differential ablation evidence agrees with the model. Although the optical data lacks altitude measurements, figures 4 and 5 also show more volatile elements are observed at a higher density earlier in time, implying the altitude relationship.

Ca lines were identified in almost all of the 22 meteors in this campaign, and Na, Fe, and Mg were observed in many meteors. Spectral observations of these meteors come close to the noise level of the camera. The diffraction grating spreads photons

from the already optically-faint, small-mass meteors, into a range of wavelengths. This causes the first order spectrum of each meteor to become faint at both extremes of the wavelength spectrum. Nearing the smaller wavelengths ($< 500$ nm), artifacts can appear from the normalization process. This is due to the increasing intensity of background radiation in the spectra which is visible in the star spectrum in Figure 3. In addition, the short exposure time of the detector ensures good temporal resolution to identify differential ablation, but sacrifices signal in each frame. For the fainter meteors, it was necessary to coadd multiple

frames in order to identify the composition.

We presented a study of optical spectra for 22 milligram-size meteors which showed evidence for differential ablation. This result pushes the observational limitations of the current camera system, using a diffraction grating to optically detect very faint meteors at high enough signals to detect spectral lines at high temporal cadence. Stronger and more varied spectral lines were found in meteors with greater brightnesses and therefore greater mass estimates, which shows that the observations of the

faintest and smallest meteors approached the noise level of the detector. These observations present a case for the feasibility of using such spectral observations of smaller mass meteors to begin to better characterize the composition of the the smaller mass (milligram-sized) meteors. As the capabilities of imaging technology increases, so will our knowledge of the smaller, more numerous meteors through targeted spectral observations.



*Data availability.* The optical and spectral data of each meteor can be accessed on Github, DOI 10.5281/zenodo.4009858. Raw data and
data without observed meteors can be obtained from the corresponding author upon request.

**Appendix A**

**A1**

*Author contributions.* Robert G. Michell and Marilia Samara designed and conducted the experiment. Robert G. Michell also reviewed,
edited, and supervised the writing and analysis. Emma R. Mirizio developed software, performed analysis, created visualizations, and pre-
pared the manuscript with contributions from coauthors.

*Competing interests.* The authors declare that they have no conflict of interest.

*Acknowledgements.* These observations were funded in part by a Southwest Research Institute internal research and development grant.
Robert G. Michell was supported by NSF Grant # AGS1523097



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
