# Peer review of "Dynamic Spectra of Small-Mass Meteors"

_Annales Geophysicae, 2020_

## Referee Comment (RC1) · Pete Gural (Referee) · 17 Oct 2020

Overview This research tried to simultaneously collect optical spectra and radar returns for faint meteors, but the use of the term faint here really corresponds to moderately bright meteors above magnitude 3.0 and no radar counterparts were observed. So the paper examines the optical spectra obtained but only at a very high level. Differential ablation was allegedly observed but no conclusive evidence was shown in the figures.

Technical Questions Section 1 44 - Explain how meteor composition helps you to constrain luminous efficiency ? 48 - How do radar head-echoes provide composition information ? Elements are not directly observed, so is this inferred based on past optical studies ? Explain better. 56 - These are elements that are efficient emitters and make themselves easily visible in meteor spectra. This is not the same as the most common elements in meteoroids and thus are only partially representative of the true elemental

abundances. Clarify that these are common spectral element emission lines in meteor spectra. 58 - Provide a reference. 61 - Borovicka's papers were some of the earliest evidence of differential ablation associated with early Na loss. He should be cited. 64 - Section 1 should have (or end) with a statement of your goal for this data collection and analysis. You bring up a lot of good points, so focus your specific project goals and objectives for the reader. Section 2 67,71 - I would not call meteors with an absolute magnitude lower limit of +3 as "faint". These are easily seen by networks of moderate field of view non-intensified cameras deployed around the world by amateurs and professionals. I understand you need the intensification because of the spectral dispersion, but you are looking at a very high mass range for the radar. 73,76 - If the spectral camera was pointed at the zenith, please explain that you must be seeing both the zeroth order and first order spectra in the FOV y using a low dispersion grating. Otherwise you would need to point the grating camera off the zenith to capture first order spectra for meteors passing through the zenith. 77 - What type of calibration. Are you referring to spectral response of the camera/lens optical system ? Or was this strictly for wavelength calibration as I see later in the paper you use a F0 star for spectral response ? So what type of lab calibration is this and what was it used for ? Predeployment system testing ? 79 - Why were meteors generally traveling east-west? Was this associated with a sporadic source location? Generally sporadic meteors move randomly. I am not aware of a meteor shower active May 17-20 that would give you that preferred radiant direction. 84 - A diagram of where in the FOV the spectral meteors passed (zeroth order) relative to the radar beam would be helpful. 84 - Do you fold in the optical observations of Mitchell 2019 at all in this paper. What is the coincidence of optical meteors detected by both the spectral camera and Mitchell 2019 optical camera. Figure 1 should more clearly point out the 1st order spectrum and the meteor (zeroth order) trace for the uninitiated reader. Figure 1 - From the images, you obviously have other stars that you could have used and done a smoother averaging of the spectral response of the system. Why limit it to one star ? Figure 2. It not good practice to use only 4 stars in a photometric calibration fit. You should do this

for as many stars as possible and spread over the night to obtain a more reliable and believable fit. Figure 4 - I see only 3 frames but the caption states 6 frames Figure 4 and 5 - These spectra look like noise and anyone would be hard pressed to say they see spectral emission lines evident. Perhaps better to show the temporally integrated spectrum since Figure 1 does show clear evidence of a meteor spectrum. For the temporal evolution, try combining 2 or 3 adjacent frames to get the spectral lines to show more clearly. 131 - There is no evidence for differential ablation shown in any figure. Is this based on visually looking at the spectral images or examining spectral wavelength plots like figure 4 and 5 ? 133 - I am highly suspect of Ca being visible in these meteors. Usually this element is associated with very bright fast moving meteors. Figure 3 left panel shows virtually no spectral response of the system to the F0 star down at 380 nm. So how can you detect Ca so readily - is the camera a blue sensitive intensifier. But that still doesn't jive with the star spectral response curve in Figure 3 left where the star's black body spectra peaks near 380 nm (Figure 3 right) but the response is at the noise level. Point out the magnitude of the F0 star. 135 - What background radiation are you referring to ? Are you simply talking about background noise and the spectra is buried in the noise ? Overall - you need to show a few actual spectral IMAGES (zoomed in) with lines labeled and then associated integrated line plots along the meteor trace. Otherwise you show no visible evidence of the conclusions stated in the end. Did you try to do an abundance ratio between the lines? Did you try to associate any meteor direction to an active shower those night's - how do you conclude that they are mostly sporadic for this bright a magnitude range ? Explain in the summary conclusions what would you do differently next time to improve the system and analysis. For example... triangulation of the meteors to get true velocities and heights. Did you do any flat fielding or dark removal. It is not stated specifically if that was done anywhere in the paper, but seemed to be implied by the caption of Figure 1.

Editing Issues None found
* * *

---

## Short Comment (SC1) · 29 Oct 2020

Review report "Dynamic Spectra of Small-Mass Meteors" by Emma R. Mirizio, Robert G. Michell, and Marilia Samara. submitted to Annales Geophysicae

This article presents the results of a spectroscopic observations of faint meteors made from Arecibo during an observing campaign in May 2012. The scientific interest is reduced because the authors have not clearly identified the origin of the meteors studied, and the paper lacks of clear goals, more than a presentation of results. I also found that the paper lacks of scientific justification for many of the results presented. On the other hand, the manuscript doesn't include a proper citation of previous literature on the topic. Finally, the scientific discussion and the conclusions are too vague as well. In these circumstances, I think that the authors should rewrite the entire manuscript. In order to help them to make it publishable, let me provide a detailed review with major

and minor issues.

First of all, the abstract and the introduction should be significantly improved. I found that some of the abstract statements are too optimistic, and does not correspond to well-sustained findings. Meteor spectra presented in this manuscript have very low resolution, according the very few examples provided (e.g. Fig. 1). In addition, some assessments are too vague. You are not presenting comparisons to demonstrate differential ablation, neither the results seem to suppose "a greater understanding of the composition" of small meteoroids.

The stellar calibration is well done although perhaps the number of stars could be too small in some cases (this should be explained case by case). In addition, I found that the authors are not saying anything about an additional key correction needed. The meteors are moving in the CCD detector at a very different angular velocity than the stars that for these short videos appear static (see e.g. Rendtel, 1993). In consequence, the meteor magnitudes compared to the stars directly are systematically understimated. Our experience indicates that such loss could be so high as 3-4 magnitudes, depending of observing circumstances.

In consequence, if the authors want to infer realistic meteoroid masses, the meteor velocity must be properly quantified, and a correction applied to the stellar comparison in order to get the apparent meteor magnitudes. A case study should be selected and entirely reduced in the results section, before the discussion. In addition, if double station was performed and the distance to the observing station is known, the authors could estimate the absolute magnitude of the meteors at a distance of 100 km. All these details should be provided in a specific table for each meteor, perhaps as an annex.

Concerning the main results, I think that the presentation and discussion of the most relevant ones concerning the bulk chemical elemental variations are not properly made. First of all, the authors need to quantify the spectral resolution (e.g. in nm/pixel) and

discuss properly if the spectra contain blended lines (e.g. doublets) and how this affects the results. Absolute calibration of the spectra is not possible for low-resolution spectra without a reconstructed trajectory. A thing that the authors can do is to quantify in each meteor spectrum the maximum intensity of the main emission lines. Then, the results can be presented properly in a ternary diagram showing the main rock-forming elements: Mg, Fe, Ca, etc... (see e.g. Madiedo et al., 2013).

In reference with Table 1, I found that a clear identification of the plausible origin of the meteoroids in missing. The authors should be able to identify if each meteor can be associated with a stream or being sporadic in nature. In the discussion they should also note that the observed chemical variations could be stochastic in nature, given the small sizes so the proportions of minerals could be highly variable (read e.g. Rietmeijer, 2004).

I also found that the authors should revise much more papers from scientific literature that are relevant to discuss these results. Small meteoroids are made by fine-grained aggregates that are built by diverse mineral components that end in low tensile strengths (Blum et al., 2006; Trigo-Rodríguez and Blum, 2009). The random distribution of such mixtures might produce different bulk chemical compositions, and tensile strengths (Rietmeijer, 2004; Trigo-Rodríguez and Blum, 2009). Some moderately volatile elements like e.g. Na are preferentially depleted in space during close approaches to the Sun or during long exposures to interplanetary medium (Trigo-Rodríguez et al., 2004).

The spectral results should be properly presented and compared with previous work of faint video meteors. I must say that the article is deeply biased concerning citations, and they are many papers in scientific literature discussing video spectra that might be relevant to improve the scientific discussion of the results (e.g. Borovička J., 2001; Koten et al., 2008; Vojáček, V. et al., 2019). All these papers could give you useful clues to deal with the reduction of video spectra of faint meteors and obtain valuable chemical clues on their bulk chemical compositions. In addition, you could made a

better introduction about the use of video sensors for spectroscopy.

At the end of the paper I found that you should separate the discussion from the conclusions. In fact, you should clearly state as separate points the main findings of this research to demonstrate us that it deserves to be published.

Minor things:

1) The terminology should be applied correctly. For example, in the abstract the authors state "small-mass (2-200 mg) meteors", but they should write "faint meteors produced by small-mass meteoroids". Meteor brightness depends on luminous efficiency that is not studied here, so I wonder how they estimate such an accurate mass distribution. 2) Meteor spectra presented in this manuscript have very low resolution. Following previous comments, the authors should state this is the abstract and related sections. Some average values should be better presented and discussed (ternary diagrams) 3) Fragmentation is not explained in the introduction despite that might play a role in the ablation of stream meteoroids (Ceplecha et al., 1993) 4) Table 1 present quite surprising values. For example an extremely rare meteoroid penetrating at 77 km/s (then, from interstellar origin). Please explain the accuracy of the data, and your method to infer velocities. Can you really get for so low resolution imagery an uncertainty of 0.1 km/s? Please revise and choose an example case in the paper, explaining the trajectory reconstruction, and the method to get masses, azimuth angles, and velocities. Some of these values seem to be too precise, so need to be properly justified. 5) For the very fast meteors, have the authors detected the presence of 2nd order ionized emission lines? In positive case, have these lines influence in the meteor magnitude (luminous efficiency)? 6) In page 2 and in the discussion section the concept of differential ablation appears. This should be properly explained with clear citation to the papers describing this physical process (Borovička, 1994; Trigo-Rodríguez et al., 2004).

In consequence, I think that this manuscript should be properly revised before being considered for publication. Then, my recommendation is a major review to satisfy all

the above mentioned points.

Additional references

Abe S. et al. (2004) Video and Photographic Spectroscopy of 1998 and 2001 Leonid Persistent Trains from 300 TO 930 nm. Earth, Moon, and Planets 95, 265-277.

Blum J. et al. (2006) The physics of protoplanetesimal dust agglomerates. I. Mechanical properties and relations to primitive bodies in the solar system, Astrophysical Journal 652, 1768-1781.

Borovička J. (2001) Video spectra of Leonids and other meteors. Proceedings of the Meteoroids 2001 Conference, 6 - 10 August 2001, Kiruna, Sweden. Ed.: Barbara Warmbein. ESA SP-495, Noordwijk: ESA Publications Division, ISBN 92-9092-805-0, 2001, p. 203 – 208

Borovička, J. (1994) Two components in meteor spectra. Planetary and Space Science 42, 145-150.

Ceplecha, Z. et al. (1993) Atmospheric fragmentation of meteoroids. Astronomy and Astrophysics 279, no. 2, p. 615-626

Koten P. et al. (2008) Video Observations of the 2006 Leonid Outburst. Earth, Moon, and Planets, Volume 102, Issue 1-4, pp. 151-156.

Madiedo J.M., et al. (2013) The 2011 October Draconids outburst. II. Meteoroid chemical abundances from fireball spectroscopy, Mon. Not. Royal Astron. Soc. 433, 571-580.

Rendtel J. (1993) Handbook for photographic meteor observations. Handbook for photographic meteor observations., by Rendtel, J.. IMO Monogr., No. 3

Rietmeijer, F.J.M. (2004) Interplanetary dust and carbonaceous meteorites: constraints on porosity, mineralogy, and chemistry of meteors from rubble-pile planetesimals. Earth, Moon & Planets 95, 321-338.
Trigo-Rodríguez J.M. (2019) The flux of meteoroids over time: meteor emission spectroscopy and the delivery of volatiles and chondritic materials to Earth. In Hypersonic Meteoroid Entry Physics, Colonna G., Capitelli M. and Laricchiuta A. (eds.), Institute of Physics Publishing, IOP Series in Plasma Physics, pp. 4-1/4-23.

Trigo-Rodríguez J.M. and J. Blum (2009) Tensile strength as an indicator of the degree of primitiveness of undifferentiated bodies, Planetary and Space Science 57, 243-249.

Trigo-Rodríguez J.M., J. Llorca and J. Fabregat (2004) Chemical abundances determined from meteor spectra: II. Evidence for enlarged sodium abundances in meteoroids, Montly Notices of the Royal Astronomical Society 348, pp.802-810.

Vojáček, V. et al. (2019) Properties of small meteoroids studied by meteor video observations. Astronomy & Astrophysics, Volume 621, id.A68, 21 pp.

Please also note the supplement to this comment:
https://angeo.copernicus.org/preprints/angeo-2020-61/angeo-2020-61-SC1-supplement.pdf
* * *

---

## Author Comment (AC1) · 18 Feb 2021

The authors would like to thank the reviewer for their comments that will greatly improve the manuscript.

In response to technical questions, the authors will add explanations in the manuscript for each of the questions raised.

In terms of meteor composition's relationship to luminous efficiency, by understanding the ablation process as well as the initial proportions of the meteoroids' composition, fewer assumptions would be made in mass estimation. Particularly, the luminous efficiency can vary based on meteor composition due to bright spectral lines from a particular element (Borovicka, 1994). This would change the generally accepted values of luminous efficiency for each case rather than the wide variety of values currently used.

Radar head-echoes were used to find differential ablation in (Janches, 2009) by modeling the change in signal at different points in time. Meteor composition and differential ablation have been determined from meteor head-echoes for microgram sized meteors in the sporadic background by modeling and observing the change in radar signal over time as various elements were expected to ablate.

We will clarify the meaning of "common" and add a reference: "Commonly observed bright spectral lines in sporadic meteors correspond to elements such as Mg, Fe, Ni, and Na as well as smaller abundances of Ca (Harvey, 1973)." Added additional Borovicka references.

A statement of goals will be added: "The goal of this data collection and analysis is to find the composition and search for differential ablation for a sample of meteors using the observed spectra, as well as estimate the mass, velocity, and brightness of each meteor using optical data. In addition, this acted as a proof-of-concept study to push the limits of our imaging technology for observing the smallest meteors for which spectra could be resolved."

"Faint" will be clarified in relation to the other meteors in the sample rather than the objective quality of the meteors, and the description of the spectrum obtained from the camera pointing at zenith was clarified. Zeroth order spectrum was defined and added to Figure 1.

We discuss a separate sample of meteors from (Michell 2019). We will expand and clarify on the calibration methods using stars and lab methods in the following way. The stellar calibration resulted in a good fit, and adding additional dimmer stars to the calculation did not improve the fit or values for determining the lower bound of mass. Due to the speed at which meteors travel over the detector, it is possible that the magnitudes are underestimated when determined from the more static stars (Rendtel, 1993).

In the results and conclusion sections, we will add a figure of the image of the spectra

and greatly explain the calculations made. Fixed the typo of 6 frames to 3 frames, and clarified that the figures will include a few frames of the image added together for better SNR. This sample was not found to be associated with any meteor shower, which resulted in defining the sample as consisting of sporadic meteors. For future work, adding radar measurements or triangulation, as the referee mentioned, would greatly improve the analysis. This would allow for estimates rather than lower limits on parameters such as velocity and mass. The additional figure should clarify the conclusions.

[Figure]

[Figure]

**Fig. 1.** Image of the spectrum of the same meteor as in Figures 4 and 5 visualized in with the intensity of spectral lines as brightness deceasing over time (left) and as line plots (right).

**Example meteor and spectrum**

Meteor and
1st order
spectrum

**Fig. 2.** An example raw image of a meteor and its spectra (left) that is uncalibrated and unflattened with the zeroth order image of the meteor circled in red.

---

## Author Comment (AC2) · 18 Feb 2021

The authors would like to thank the reviewer for their comments that will greatly improve the manuscript.

The luminous efficiency can vary based on meteor composition due to bright spectral lines from a particular element (Borovicka, 1994). This would change the generally accepted values of luminous efficiency for each case rather than the wide variety of values currently used.

We will clarify the meaning of "common" and add a reference: "Commonly observed bright spectral lines in sporadic meteors correspond to elements such as Mg, Fe, Ni, and Na as well as smaller abundances of Ca (Harvey, 1973)." Added additional Borovicka references.

[Figure]

A statement of goals will be added: "The goal of this data collection and analysis is to find the composition and search for differential ablation for a sample of meteors using the observed spectra, as well as estimate the mass, velocity, and brightness of each meteor using optical data. In addition, this acted as a proof-of-concept study to push the limits of our imaging technology for observing the smallest meteors for which spectra could be resolved."

The stellar calibration resulted in a good fit, and adding additional dimmer stars to the calculation did not improve the fit or values for determining the lower bound of mass. Due to the speed at which meteors travel over the detector, it is possible that the magnitudes are underestimated when determined from the more static stars (Rendtel, 1993). This reference suggested by the referee will be added.

In the results and conclusion sections, we will add a figure of the image of the spectra and greatly explain the calculations made. This sample was not found to be associated with any meteor shower, which resulted in defining the sample as consisting of sporadic meteors. For future work, adding radar measurements or triangulation would greatly improve the analysis. This would allow for estimates rather than lower limits on parameters such as velocity and mass. The additional figure should clarify the conclusions.

From the referee's minor comments, we will clarify the language of "faint" and the resolution of the spectra. Also, additional references are included in the discussion of differential ablation. In terms of the error associated with the values in the table, due to the lack of triangulation, the velocity and mass will be limits rather than estimates. We will clarify this.

The additional references the referee listed will be further analyzed and included, such as the additional Borovicka papers, Trigo-Rodriguez papers, and Ceplecha papers.

The manuscript has been significantly reworked to better explain the process of taking the images and finding meteor composition.

[Figure]

**Fig. 1.** Image of the spectrum of the same meteor as in Figures 4 and 5 visualized in with the intensity of spectral lines as brightness deceasing over time (left) and as line plots (right).

**Example meteor and spectrum**

Meteor and
1st order
spectrum

**Fig. 2.** An example raw image of a meteor and its spectra (left) that is uncalibrated and unflat-
tened with the zeroth order image of the meteor circled in red.